# A Kinetic Study on Crystallization in TiO_2_-SiO_2_-CaO-Al_2_O_3_ Glass under Nucleation Saturation Conditions for the High Value-Added Utilization of CaO-SiO_2_-Based Solid Wastes

**DOI:** 10.3390/ma16114165

**Published:** 2023-06-02

**Authors:** Zhen Wang, Renze Xu

**Affiliations:** 1State Key Laboratory of Multiphase Complex Systems, Institute of Process Engineering, Chinese Academy of Sciences, Beijing 100190, China; 2School of Chemical Engineering, University of Chinese Academy of Sciences, Beijing 100049, China

**Keywords:** crystallization kinetics, glass, non-isothermal, slag, glass–ceramics, differential thermal analysis, TiO_2_

## Abstract

A kinetic study of the non-isothermal crystallization of CaO-SiO_2_-Al_2_O_3_-TiO_2_ glass was carried out using the Matusita–Sakka equation and differential thermal analysis. As starting materials, fine-particle glass samples (<58 µm), case defined as ‘‘nucleation saturation’’ (i.e., containing such a large number of nuclei that the nucleus number is invariable during the DTA process), became dense bulk glass–ceramics through heat treatment, demonstrating the strong heterogeneous nucleation phenomenon at the juncture of particle boundaries under “nucleation saturation” conditions. Three types of crystal phase are formed during the heat treatment process: CaSiO_3_, Ca_3_TiSi_2_(AlSiTi)_3_O_14_, and CaTiO_3_. As the TiO_2_ content increases, the main crystal shifts from CaSiO_3_ to Ca_3_TiSi_2_(AlSiTi)_3_O_14_. The *E*_G_ values (activation energy of crystal growth) are in the 286–789 kJ/mol range. With increasing TiO_2_, *E*_G_ initially decreases (the minimum appears at 14% TiO_2_), and then, increases. When added within 14%, TiO_2_ is shown to be an efficient nucleating agent that promotes the growth of wollastonite in a two-dimensional mechanism. As TiO_2_ further increases to exceed 18%, it is no longer just a nucleating agent but becomes one of the major components in the studied glass, so, in turn, it undermines the crystallization of wollastonite by forming Ti-bearing compounds, resulting in a tendency toward surface crystallization and higher activation energy of crystal growth. For glass samples with fine particles, it is important to note the “nucleation saturation” case for a better understanding of the crystallization process.

## 1. Introduction

The CaO-SiO_2_-Al_2_O_3_ oxide system has long attracted plentiful attention owing to its easy availability in the mineral utilization industry and extensive application prospects in ceramics. On the one hand, among the wide variety of by-products in the mineral smelting process, the CaO-SiO_2_-Al_2_O_3_ ternary system is of particular importance, and is ubiquitous in most metallurgical processes, such as the iron and steelmaking processes and the Al and Ca refining processes [1,2,3,4]. It is usually classified as metallurgical solid waste that causes potential and long-term harm to the environment if it is mishandled. However, these slags, in turn, are a ready source of CaO, SiO_2_, and Al_2_O_3_, which are the basic components for numerous ceramic materials, such as calcium aluminosilicate glasses, low-temperature co-firing ceramic materials, microwave dielectric materials, and glass–ceramics with propitious electrical properties, superior wear resistance, chemical resistance, high-temperature strength, dimensional stability, etc. [5,6,7,8,9,10]. The development of these new advanced ceramics is difficult without deep knowledge of the CaO-SiO_2_-Al_2_O_3_ system. Researching the CaO-SiO_2_-Al_2_O_3_ system is of great significance, both from a waste utilization perspective and an advanced ceramic material development perspective.

Whether it is the utilization of metallurgical solid waste containing CaO, SiO_2_, and Al_2_O_3_ or the exploitation of CaO-SiO_2_-Al_2_O_3_-based glass–ceramics, the study of the crystallization characteristics is important and indispensable. This is because the crystallization behavior of crystal phases directly affects the methods of valuable component separation from solid waste or the exploration of the propitious properties of glass–ceramics. Furthermore, as the solid form of liquidus metallurgical solid waste and the starting material for the preparation of glass–ceramics, glass is a critical state that deserves more attention. Thus, numerous studies focus on the crystallization behaviors of CaO-SiO_2_-Al_2_O_3_-based glass. Usually, for a particular property or performance, the types of crystal or the amount of a crystal phase often require precise control. For this purpose, one additive or more is usually introduced into the glass, acting as property modifiers or nucleating agents. Among some of the widely researched additives, such as B_2_O_3_, CuO, TiO_2_, CaF_2_, ZrO, MnO_2_, Cr_2_O_3_, etc. [11,12,13,14,15,16], TiO_2_ has been widely accepted as an important nucleating agent [14,15,16,17,18]. Most of these studies have looked at the influence of TiO_2_ on the mechanical or dielectric properties of glass ceramics [14,15,16,17], and the investigations on the kinetics of crystallization are relatively poor. First, some studies on crystallization behavior are based on metallurgical slag containing various components (such as Na_2_O, MgO, P_2_O_5_, CaF_2_, etc.) in addition to CaO, SiO_2_, Al_2_O_3_, and TiO_2_ [18,19,20,21]. These studied systems are far more complex with various types of crystal phase, and the accurate roles of small amounts of TiO_2_ in them are difficult to explore clearly, which is a disadvantage that affects our ability to understand the present relatively simple CaO-SiO_2_-Al_2_O_3_-TiO_2_ system. Second, the limited number of studies on the CaO-SiO_2_-Al_2_O_3_-TiO_2_ glass system either contain a high content of Al_2_O_3_ so that the composition of the studied system is very different from those of the metallurgical solid wastes, or suggest that TiO_2_ is not an effective nucleating agent in this glass system [22,23,24]. In general, there is still a great issue regarding the role of TiO_2_ in the crystallization of CaO-SiO_2_-Al_2_O_3_ glass, which deserves further exploration.

There has been widespread application of non-isothermal techniques to study the crystallization kinetics of glass via differential scanning calorimetry (DSC) or differential thermal analysis (DTA) in different fields [25,26,27,28]. The techniques have contributed a lot to investigations on the nucleation and crystal growth processes occurring throughout the glass transition process as it is heated, by providing important crystallization parameters such as glass transition temperature, crystallization temperature, transformation enthalpy, the crystal growth mechanism, and activation energy. There are various models or equations to help explain the non-isothermal crystallization kinetics according to DTA or DSC measurement [25,28,29,30,31,32].

However, this literature review shows that there exists some misunderstanding about the use of these equations or models. For example, because the crystal growth mechanism (i.e., bulk crystallization or surface crystallization) is not considered, the Kissinger equation is only for occasions whereby the nucleus number remains static during crystal growth [28,32,33]. However, it is still used directly by many authors to examine the crystallization kinetics of all amorphous materials, including situations whereby the number of nuclei increases obviously during the DTA process, which may lead the authors to draw misleading conclusions. Moreover, it has been found that throughout the heat treatment of the DTA test, the glass particles’ size can affect the crystal growth pattern, thereby influencing the identification of crystal growth dimensionality and the calculation of activation energy [33,34,35,36]. For example, the crystal growth for fine particles probably corresponds to a two-dimensional growth pattern which is interface reaction-controlling, while for coarse particles, it probably corresponds to a three-dimensional growth pattern, which is diffusion-controlling [33]. This is associated with a different relationship between n and m (two parameters represent the dimensionality of crystal growth). When the crystallization mechanism and activation energy are estimated based on some of these equations, the effects of particle size must be fully considered to obtain the correct relationship between n and m, thereby ensuring the correctness of calculating activation energy. However, in some studies, either the influences of particle size are ignored, or the state of “nucleation saturation” (i.e., the nucleus number is invariable during DTA process) for the fine-particle as-quenched glass samples [26,32,37] is poorly comprehended in the determination of the values of n and m; thus, some of the various rules of activation energy values proposed by these studies may be obscure, and the proposed conclusions may lead to misunderstandings. Another work published by our team on the crystallization kinetics of CaO-SiO_2_-B_2_O_3_-TiO_2_ glass takes into account the occurrence of “nucleation saturation”, and confirms that it is necessary to keep the effects of glass particle size in focus to correctly calculate the kinetic parameters [38].

Therefore, to sum up, much more work needs to be conducted to research the non-isothermal crystallization of TiO_2_-SiO_2_-CaO-Al_2_O_3_ as-quenched glass with full consideration of “nucleation saturation”. This work aims to investigate the influences of TiO_2_ on the crystal growth mechanism and crystallization activation energy of CaO-SiO_2_-Al_2_O_3_ glass in the case of “nucleation saturation”, providing a scientific and accurate guide for the association of the high-value-added utilization of metallurgical slag and the development of high-performance ceramics.

## 2. Materials and Methods

### 2.1. Glass Preparation

Based on the common composition of some metallurgical slag that is an important source of TiO_2_ and the amount of TiO_2_ in ceramic materials, the chemical compositions of the investigated samples were identified and are presented in Table 1. CaCO_3_, TiO_2_, Al_2_O_3_, and SiO_2_ served as raw materials to prepare the starting samples. All chemicals were of analytical grade. The impossibility of accurately weighing CaO (CaO + H_2_O (moisture in the air) = Ca(OH)_2_) made CaCO_3_ an alternative. During the following preparation of glass samples, the temperature gradually rose to higher than 900 °C, and CaO was formed from the decomposition reaction of CaCO_3_. According to Table 1, each raw material’s mass was calculated and weighed with a precision of 0.0001 g, and then, the four raw materials of each starting sample were ground and mixed well.

The melting–quenching method was used to prepare glass samples. Each of the four starting samples was heated to 1550 °C in an air atmosphere using a platinum crucible as the melting reactor. The temperature was held at 1550 °C for two hours to give the starting sample enough time to melt completely and form a homogeneous liquid. Then, each of the melted starting samples was thrown into ice water to be quenched. The quenched samples were the so-called glass samples, which were confirmed via X-ray diffraction (XRD, model: X’Pert PRO MPD, PANalytical B.V., Almelo, The Netherlands).

### 2.2. Test of Differential Thermal Analysis

Each glass sample was pulverized and sieved to obtain glass powder (30–58 µm) for DTA measurement. A DTG-60H (SHIMADZU, Kyoto, Japan) calorimeter was used to finish the DTA measurements. A platinum crucible was used as the reactor during the DTA measurements. During the non-isothermal heat treatments, each powdered glass sample was heated from room temperature to 1500 °C across the glass transition temperature (T_g_). Each sample was set to undergo 5 °C/min, 10 °C/min, 20 °C/min and 30 °C/min heating rates, respectively. During the DTA test, Al_2_O_3_ was used as the reference material, and Ar (99.999%) as a shielding gas.

### 2.3. Crystal Identification and Phase Analysis

Each powdered glass sample was set to undergo the same heat treatment process as the DTA test in a routine furnace to prepare the corresponding crystallized samples for further identification of the crystals. The above obtained DTA curves of 10 °C/min were used as the reference. Each powdered glass sample was heated from 30 °C to the end temperature of the exothermic peak, and then, quenched by air. The heating rate of this process was 10 °C/min. The crystals in these heat-treated samples were tested first via XRD. The morphology and phase composition were tested using a scanning electron microscope (SEM) and an energy dispersive X-ray spectrometer (EDS). All of the crystallized samples were divided into two parts. One part was ground into powder to undergo XRD measurement using an X’Pert PRO MPD X-ray diffractometer. The other part was used to prepare the SEM sample by being embedded with resin and polished to expose the cross-section. After being coated with Pt, this sample was used to perform an SEM-EDS test to analyze the crystal phases’ composition and morphology. The SEM-EDS tests were carried out using a Gemini 300 microscope manufactured by Zeiss matching an EDS (model: Ultim MAX, OXFORD).

## 3. Results and Discussion

### 3.1. Crystal Identification

Figure 1 displays the XRD patterns of the quenched samples, showing their glassy nature. The absence of a crystal diffraction peak indicates that the samples were liquid at 1550 °C.

Figure 2 gives the XRD results for the corresponding crystallized samples of the four designed glass samples. It shows that three different crystals exist in these crystallized samples: wollastonite (CaSiO_3_), perovskite (CaTiO_3_), and calcium aluminum silicon titanium oxide (Ca_3_TiSi_2_(AlSiTi)_3_O_14_). As the addition of TiO_2_ changes from 10% to 22%, the peak intensity of Ca_3_TiSi_2_(AlSiTi)_3_O_14_ decreases first, and then, increases, yet that of CaTiO_3_ and CaSiO_3_ shows the reverse change trend, indicating that the crystallization of Ca_3_TiSi_2_(AlSiTi)_3_O_14_ becomes weak first, and then, heightens; however, the crystallization of CaSiO_3_ and CaTiO_3_ increases first and then decreases.

Figure 3 presents the crystal morphology observed via SEM-BSE and the crystal phase identification analyzed via SEM-EDS for the S1, 2, 3, and 4 crystallized samples. Consistent with the XRD results, the same three crystalline phases are identified via morphology observation and EDS analysis. It can be determined from the amount of each crystalline phase in Figure 3 that the amount of Ca_3_TiSi_2_(AlSiTi)_3_O_14_ decreases with the changing of TiO_2_ from 10% to 14%, while as the content of TiO_2_ increases to 18% and 22%, the amount of Ca_3_TiSi_2_(AlSiTi)_3_O_14_ increases obviously, and Ca_3_TiSi_2_(AlSiTi)_3_O_14_ displaces CaSiO_3_ as the predominant crystalline phase. The crystallization amount of CaSiO_3_ is just the opposite.

In addition, although the starting materials for the DTA measurements were fine-particle glass (<58 µm), bulk glass–ceramic samples were produced after heat treatment in this work (Figure 4). From the cross-section of the bulk glass–ceramic samples in Figure 3, it can be observed that the inner of the bulk glass–ceramic samples is dense, crevice-free, and nonporous. This demonstrates the phenomenon of strong heterogeneous nucleation at the junction of particle boundaries in the case of “nucleation saturation”. In Figure 3, the size of the Ca_3_TiSi_2_(AlSiTi)_3_O_14_ and CaSiO_3_ grains is several microns, and that of the dendritic CaTiO_3_ grain is much smaller at under 1 µm. The increase in TiO_2_ content shows no marked effect on the size of the crystal phases.

Overall, the XRD and SEM results are well-matched. Upon combining these results, it can be concluded that the main crystal is CaSiO_3_ with an amount of TiO_2_ within 10~14%, and as TiO_2_ increases to 18~22%, Ca_3_TiSi_2_(AlSiTi)_3_O_14_ becomes the predominant crystalline phase. The growth of the CaTiO_3_ crystal particle is difficult, so the crystalline phase is dendritic.

### 3.2. Crystallization Kinetics

#### 3.2.1. Theoretical Basis

Many authors have published various methods to explain the non-isothermal crystallization kinetics of glass [25,26,28,29,39,40,41]. Among these, Matusita et al. underlined that taking the crystallization mechanism into account is necessary to obtain the valid activation energy [28,39,40]. The Matusita–Sakka equation published by them has been well recognized for dealing with the crystallization kinetics of glass in non-isothermal cases. It is as follows [28]:(1)$$\ln\left[-\ln\left(1-\alpha \right)\right]=-\mathrm{nln}\beta -1.052\frac{mE_{G}}{RT}+\mathrm{const}$$
where *α* is volume fraction of the formed phase at a given temperature obtained according to the crystallization peak in the DTA curves, and the calculation method of *α* was shown in another work by our team [37]; n is the Avrami exponent, and n, m are constants, representing the crystal growth mechanism and growth dimensionality; *β* is the heating rate of the DTA measurement; *T* is the absolute temperature; *R* is the universal gas constant; and *E*_G_ represents the activation energy of crystal growth.

In Equation (1), the m*E*_G_/*R* value can be determined using the following equation at a fixed heating rate:(2)$${\left.\frac{d\left(\ln\left[-\ln\left(1-\alpha \right)\right]\right)}{d\left(\frac{1}{T}\right)}\right|}_{\beta }=-1.052\frac{mE_{G}}{R}$$

According to Matusita et al. [28], the value of m*E*_G_/n can be calculated at a given α using the following equation:(3)$${\left.\frac{d\left(\ln\beta \right)}{d\left(\frac{1}{T}\right)}\right|}_{\alpha }=-1.052\frac{m}{n}\frac{E_{G}}{R}$$

#### 3.2.2. Kinetic Analysis of the Crystallization of CaO-SiO_2_-Al_2_O_3_-TiO_2_ Glass

Figure 5 presents the measured DTA curves at 5, 10, 20, and 30 °C/min for S1–S4 glass samples with different TiO_2_ contents. It can be seen that the exothermic peak caused by crystal growth appears approximately in the range of 900 °C to 1100 °C, and with the increase in heating rate, it shifts toward a higher temperature. Additionally, with the increase in TiO_2_, the temperature range of the exothermic peak shifts toward a lower temperature.

Based on the exothermic peaks in Figure 5 and Equation (3), ln*β* vs. 1000/*T* curves were drawn at different values of *α* (*α* = 0.1, 0.2, 0.3, 0.4, 0.5, 0.6, 0.7, 0.8, and 0.9 were chosen in this study) to calculate the m*E*_G_/n values. Figure 6 shows these plots. The drawing procedure of Figure 6 is as follows: first, the integral area (A) of the exothermal peak at a certain heating rate is calculated from the onset temperature to the end temperature of the exothermic peak. Then, 0.1 × A (i.e., α = 0.1), 0.2 × A (i.e., α = 0.2), …, 0.9 × A (i.e., α = 0.9) can be calculated, respectively. Additionally, each temperature, corresponding to α = 0.1, α = 0.2, …, α = 0.9, respectively, can be calculated or read from the above integral curve of the exothermal peak. Finally, the correspondence between α and temperature at different heating rates is built, and the plots are obtained. Additionally, the m*E*_G_/n can be determined by averaging the slopes of the linear fitting equations for different values of *α*. Curves of ln[−ln(1 − *α*)] vs. (1000/T) are drawn at heating rates of 5 °C/min, 10 °C/min, 20 °C/min, and 30 °C/min, respectively, to calculate the values of m*E*_G_ at different heating rates based on the equation (2), as shown in Figure 7. Additionally, the slope of the linear fitting equation is used to calculate the m*E*_G_ at different heating rates.

Dividing the values of m*E*_G_ obtained in Figure 7 by m*E*_G_/n obtained in Figure 6 provides the values of n at different heating rates. Then, the value of m can be determined according to the relationship of m and n. Thereby, the activation energies for crystal growth (*E*_G_) and the crystallization mechanism can be obtained. Generally, the relationship between n and m changes with variations in the particle size of the glass samples for DTA investigation. For instance, in a well-nucleated bulk glass, the amount of existing nuclei in the sample is so large that it does not grow anymore throughout the DTA test, so in such situations, m = n; however, in an as-quenched bulk glass, nucleation is an ongoing process through the heat treatment, and in this situation, m = n − 1 [28,33,41]. However, some important caveats are particularly noteworthy: for fine-particle glass, because of the precedence of nucleation at the particle edge, surface, corners, or junction of particle boundaries, the tremendous number of grain boundaries and the large surface area provide sufficient nucleation sites such that this glass shows the characteristic of heterogeneous nucleation, i.e., “nucleation saturation” [26,32,33,35]. Thus, for fine-particle glass, the nucleation sites on the grain surface are basically saturated, to the extent that the newly formed nuclei inside the particle, through the DTA test, can be safely neglected. This implies that the nucleus number in the fine-particle glass is supposedly constant through the DTA test, although this glass is an as-quenched glass without any nucleation treatment [26,32,33,35]. Therefore, it is suggested m = n for fine-particle glass. The starting materials employed for the DTA test are particle glass with a grain size of less than 58 µm in this work, and can thus be classified as fine-particle starting materials based on references [32,33,37]. In conclusion, m = n is proposed in this work. To sum up, the obtained crystallization parameters (m, n, m*E*_G_, and *E*_G_) and the crystal phases for each sample are presented in Table 2.

The crystal growth mechanism of the studied glass can be revealed from m and n in Table 2. Some authors [28,33,41] have proposed that for glass samples with a constant number of nuclei across the DTA run, i.e., those undergoing “nucleation saturation”, m = n = 3, 2, or 1. Here, m = n = 3, 2, or 1 represent different crystallization mechanisms: m = n = 1, surface crystallization or the crystal growth is one-dimensional; m = n = 2, the crystal growth is two-dimensional; m = n = 3, the crystal growth is three-dimensional. From the results in Table 2, it can be deduced that in this studied glass, as TiO_2_ ranges from 10 to 18%, the crystal growth corresponds to a two-dimensional mechanism; when the TiO_2_ content is 22%, surface crystallization or a one-dimensional crystal growth mechanism appears. For further confirmation of surface crystallization or the one-dimensional growth mechanism, the crystallized S4 sample was subjected to SEM to observe the crystal growth mechanism. Figure 8 presents the observed results. It is discovered that a lot of columnar crystal grains form near the sample’s surface and are nearly perpendicular to the surface. This phenomenon matches McMillan’s assertions on surface crystallization, namely, crystals grow from the surface of glass particles inward, and usually, they are oriented at 90° to the surface [42]. Moreover, it can also be seen that there are some other crystal grains inner beyond the above characteristics that may correspond to the one-dimensional growth mechanism according to the values of m and n. In general, it is suggested that the crystallization processes of the glass containing 22% TiO_2_ undergo surface crystallization and one-dimensional growth simultaneously.

Strictly speaking, the effective activation energy reflecting the kinetic barrier for crystallization should involve both the nucleation and crystal growth processes, rather than being specific to just one of the two processes. However, in this work, the occurrence of “nucleation saturation” is of concern, so the nucleation process is negligible according to the DTA test. Thereby, the activation energy for the nucleation process is negligible, and the activation energy for crystal growth, *E*_G_, is regarded as the activation energy for the whole crystallization process to measure how difficult or easy crystallization is in the studied glass. From this, the *E*_G_ values listed in Table 2 are the main parameters for investigating the relationship between crystallization capacity and TiO_2_ content. Actually, when Equation (1) is derived by Matusita et al., m and n are specially introduced into the derivation process to make the equation more precise and valid in obtaining the activation energy for crystal growth both in a quenched glass containing no nuclei and a glass containing a sufficiently large number of nuclei.

According to the results in Table 2, the changes in *E*_G_ with increased TiO_2_ are plotted in Figure 9. It is found that the values of *E*_G_ decrease initially with the increase in TiO_2_ content (reaching a minimum at 14% TiO_2_), and then, increase. With the SEM and XRD results together, it is deduced that when the TiO_2_ content increases from 10% to 14%, the glass, with CaSiO_3_ being the main crystal, shows an increasing crystallization ability. Meanwhile, as the predominant crystalline phase changes to Ca_3_TiSi_2_(AlSiTi)_3_O_14_ with TiO_2_ increasing to 18% and 22%, not only does the crystallization capacity exhibit a gradually declining trend, but the values of *E*_G_ are also higher than those of CaSiO_3_ being the main crystalline phase.

It is known that CaSiO_3_ consists primarily of the chain structural unit ${\mathrm{SiO}}_{3}^{2-}$, while the formation of Ca_3_TiSi_2_(AlSiTi)_3_O_14_ needs more complex structural units, such as ${\mathrm{Si}}_{2}O_{5}^{2-}$ coupled with Ti-O-Ti and Al-O-Al in a network structure. That is, the formation of CaSiO_3_ only involves the motion of one type of silica structural unit, while the formation of Ca_3_TiSi_2_(AlSiTi)_3_O_14_ involves the encounters, interactions, and motion of at least three types of structural unit to form a complex network structure. This means that the formation of Ca_3_TiSi_2_(AlSiTi)_3_O_14_ is much more difficult than that of CaSiO_3_. Hence, it is not difficult to understand why the crystal growth activation energies for samples in which Ca_3_TiSi_2_(AlSiTi)_3_O_14_ is the predominant crystalline phase are higher. Notably, without considering the occurrence of “nucleation saturation”, m = n − 1 would be concluded, which would lead to misinterpretation, whereby we would assume higher activation energy for CaSiO_3_ growth than for Ca_3_TiSi_2_(AlSiTi)_3_O_14_. This, obviously, is inappropriate. Thus, for glass samples with fine particles, considering the occurrence of “nucleation saturation” is necessary to help to grasp the crystallization process correctly.

Regarding the changes in activation energies with the increase in TiO_2_, on the one hand, from the SEM and XRD analysis, it can be deduced that as the content of TiO_2_ is in the range of 10–14%, the increased TiO_2_ mainly participates to form more CaTiO_3_, which competes with Ca_3_TiSi_2_(AlSiTi)_3_O_14_ for CaO, thereby promoting the disintegration of Ca_3_TiSi_2_(AlSiTi)_3_O_14_. Therefore, more SiO_2_ is released from the disintegration of Ca_3_TiSi_2_(AlSiTi)_3_O_14_ to form CaSiO_3_, thereby increasing the crystallization ability of CaSiO_3_. As the predominant crystalline phase, the competitiveness of CaSiO_3_ crystallization directly determines the crystallization ability of the glass system studied. Thus, with the content of TiO_2_ increasing from 10% to 14%, the crystallization capacity of the studied glass system presents an enhanced trend, which is consistent with the crystallization ability of CaSiO_3_.

On the other hand, as TiO_2_ reaches 18% and 22%, the added TiO_2_ and the decrease in the crystallization amount of CaTiO_3_ provide sufficient TiO_2_ to strengthen the interactions of the Si-O-Si and Ti-O-Ti network structure to form more complex structural units, thereby facilitating the formation of more Ca_3_TiSi_2_(AlSiTi)_3_O_14_ and making it the predominant crystalline phase. Another work published by our team about a structural study of the same CaO-SiO_2_-Al_2_O_3_-TiO_2_ glass shows that increasing TiO_2_ reduces the molar ratio of ${\mathrm{SiO}}_{3}^{2-}$ to a sheet structural unit. That is, increasing TiO_2_ content promotes the transformation of ${\mathrm{SiO}}_{3}^{2-}$ to a (Si, Ti) coupling complex structural unit, so the crystallization amount of CaSiO_3_ decreases and that of Ca_3_TiSi_2_(AlSiTi)_3_O_14_ increases so that it becomes the predominant phase. However, this is not matched by a corresponding enhancement in crystallization ability. Firstly, it is known that crystal growth changes from a two-dimensional mechanism to a surface crystallization mechanism or a one-dimensional mechanism as TiO_2_ increases to 22%. Additionally, the latter mechanism means a more difficult process for crystal growth, so a change in the crystal growth mechanism leads to an increase in the *E*_G_ value and a decrease in crystallization ability. Secondly, to identify the sequence of crystallization of different phases for samples in which Ca_3_TiSi_2_(AlSiTi)_3_O_14_ is the predominant crystalline phase, the S3 glass sample was used to undergo a heat treatment according to the DTA result obtained at 10 °C/min. Figure 10 gives the XRD patterns of the crystallized samples obtained via quenching at different temperatures. Figure 10 shows that at 922 °C (slightly higher than the onset temperature of the exothermic peak), at the beginning of crystal growth, there are two crystalline phases (CaTiO_3_ and Ca_3_TiSi_2_(AlSiTi)_3_O_14_), and CaTiO_3_ is the main one. As the quenching temperature increases to 952 °C (the volume fraction of crystals is nearly 0.2), CaSiO_3_ appears, and the amount of Ca_3_TiSi_2_(AlSiTi)_3_O_14_ increases obviously. It can be inferred from these results that the sequence of crystallization is CaTiO_3_, Ca_3_TiSi_2_(AlSiTi)_3_O_14_, and CaSiO_3_. The first precipitated phase has a great impact on the crystallization process because it occupies most of the initial nuclei, and in turn, its formation can supply new nuclei for the subsequently crystallized phase. So, as the second crystallized phase, the growth of Ca_3_TiSi_2_(AlSiTi)_3_O_14_ is affected by the crystallization of CaTiO_3_. As the content of TiO_2_ increases to 18% and 22%, the crystallization amount of CaTiO_3_ decreases gradually, and this may cause a disadvantageous impact on the growth of the second crystallized phase, Ca_3_TiSi_2_(AlSiTi)_3_O_14_. So, the activation energies increase, and the crystallization ability of the system is weakened.

It is reported that the crystallization mechanism of wollastonite glass–ceramic is a kind of surface crystallization [43,44]. Therefore, to sum up, the addition of TiO_2_ within a certain amount (in this case, no more than 14%) is shown to be an effective nucleating agent in this present system, not only promoting changes in the crystallization mechanism of wollastonite into two-dimensional crystallization (one form of bulk crystallization), but also improving the crystal growth ability of wollastonite in the system. Nevertheless, the further increase in TiO_2_ (in this case, exceeding 18%) makes it become a major component in the glass rather than a simple nucleating agent, so it is more likely to react with other components to form Ti-bearing compounds, thus undermining the crystallization of wollastonite. Eventually, in turn, it makes the glass tend toward surface crystallization and higher crystal growth activation energy.

## 4. Conclusions

Crystallization kinetics were investigated in CaO-SiO_2_-Al_2_O_3_-TiO_2_ glass via DTA under non-isothermal conditions. By factoring ‘‘nucleation saturation’’ into the estimation of crystal growth dimensionality, the Matusita–Sakka equation was employed to calculate the parameters of crystallization kinetics in this system. The following conclusions were drawn:

As TiO_2_ increases, the crystallization tendency of wollastonite increases first, and then, decreases, with a turning point of 14% TiO_2_. The crystallization of perovskite shows a similar trend to wollastonite, while calcium aluminum silicon titanium oxide shows an obviously opposite trend. The predominant crystal varies from wollastonite to calcium aluminum silicon titanium oxide with an increase in TiO_2_. In the case of fine glass powder as a starting material, dense bulk glass–ceramics form after undergoing heat treatment, demonstrating the fact that in the case of “nucleation saturation”, the junction of particle boundaries contains sufficient heterogeneous nucleation sites.

For samples of 10–18% TiO_2_, a two-dimensional growth mechanism is suggested for the crystallization process, while for samples with 22% TiO_2_, one-dimensional crystal growth and surface crystallization appear. The *E*_G_ values decrease initially (reaching a minimum at 14% TiO_2_) with the increase in TiO_2_ content, and then, increase. These facts indicate that the addition of TiO_2_ within a certain amount (in this case, no more than 14%) is shown to be an effective nucleating agent in this present system, not only promoting the crystallization mechanism of wollastonite in a two-dimensional form, but also improving the crystal growth ability of wollastonite in the system, while a further increase in TiO_2_ addition (in this case, exceeding 18%) makes it more likely to react with other components to form Ti-bearing compounds and undermines the crystallization of wollastonite; thus, in turn, it makes the glass tend toward surface crystallization and higher crystal growth activation energy.

When crystallization kinetics is studied under non-isothermal conditions using differential thermal analysis, for glass samples with fine particles, taking the occurrence of “nucleation saturation” into consideration is necessary to accurately determine the changes in crystal growth activation energy and understand the crystallization process correctly.

## Figures and Tables

**Figure 1 materials-16-04165-f001:**
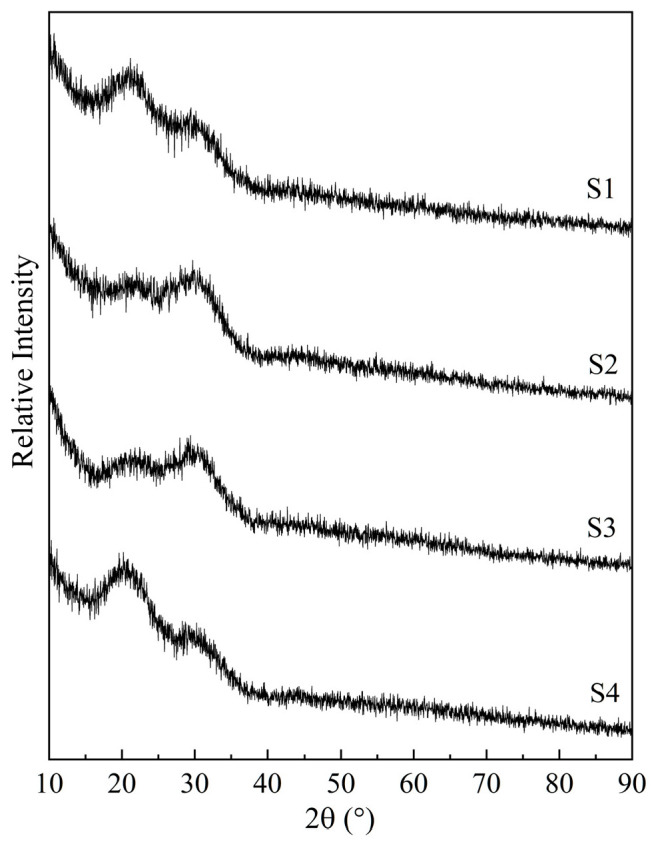
XRD results for the obtained glass samples obtained via quenching at 1550 °C. S1–S4 correspond to the sample numbers in Table 1.

**Figure 2 materials-16-04165-f002:**
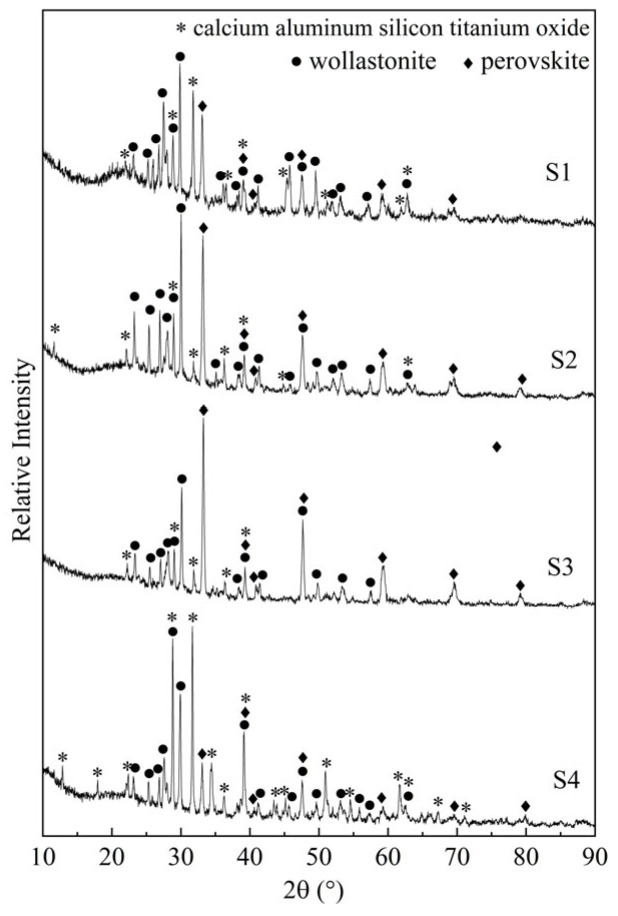
XRD phase identification for the S1–S4 crystallized samples.

**Figure 3 materials-16-04165-f003:**
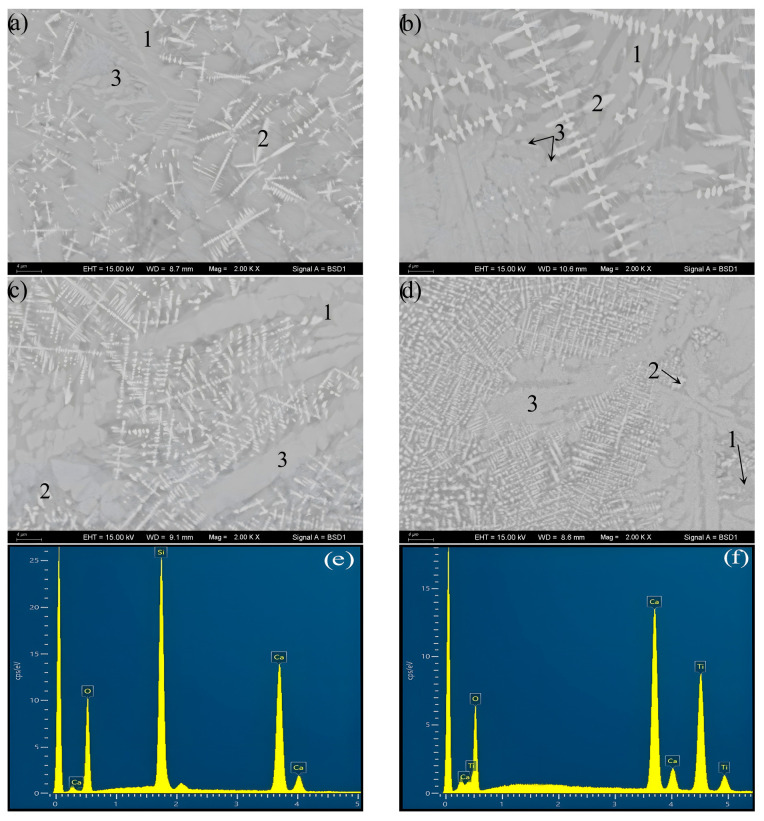
The SEM-BSE morphology observation and SEM-EDS phase identification of the S1, S2, S3, and S4 crystallized samples: (**a**) S1–10%TiO_2_, (**b**) S2–14%TiO_2_, (**c**) S3–18%TiO_2_, (**d**) S4–22%TiO_2_, (**e**) EDS analysis for phase 1, (**f**) EDS analysis for phase 2, (**g**) EDS analysis for phase 3; 1 represents CaSiO_3_ phase, 2 represents CaTiO_3_ phase, 3 represents Ca_3_TiSi_2_(AlSiTi)_3_O_14_ phase.

**Figure 4 materials-16-04165-f004:**
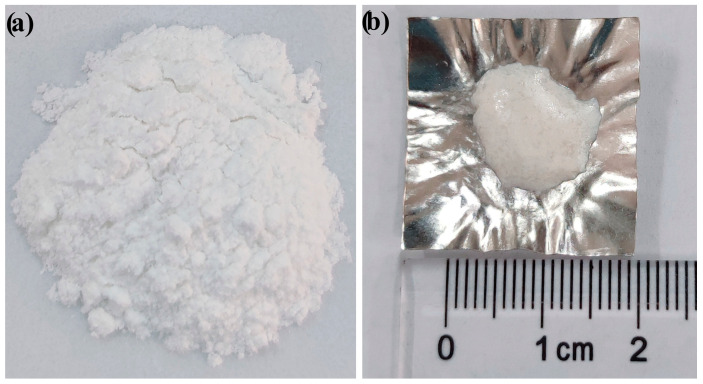
Appearance of the starting material and the heat treatment product: (**a**) glass powder (<58 µm), (**b**) the obtained bulk glass–ceramics.

**Figure 5 materials-16-04165-f005:**
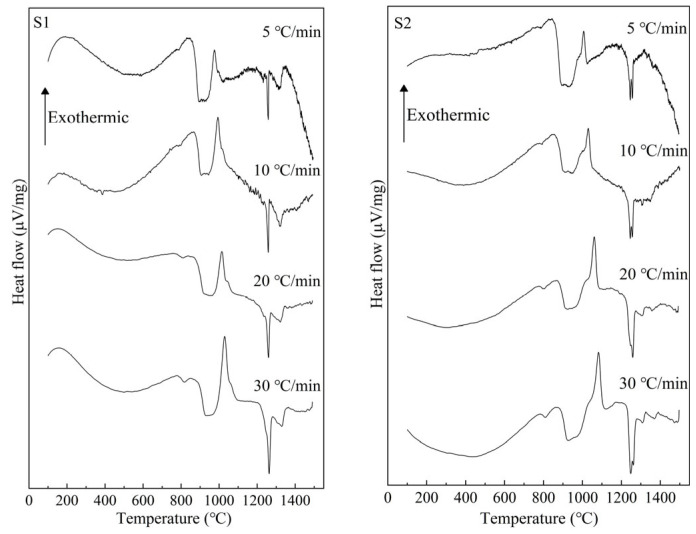
DTA measurement results for S1–S4 glass.

**Figure 6 materials-16-04165-f006:**
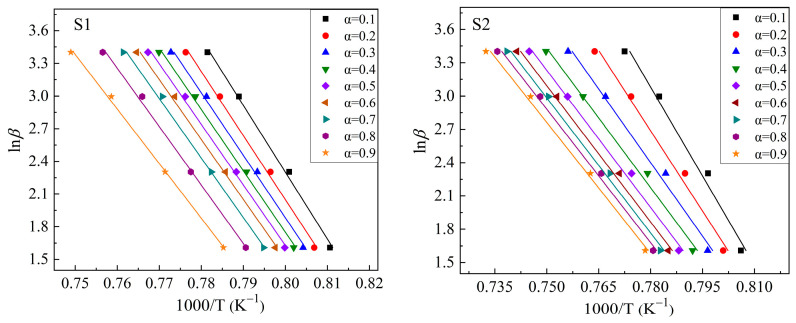
Plots of ln*β* ~1000/T for S1–S4 glass samples to calculate m*E*_G_/n values.

**Figure 7 materials-16-04165-f007:**
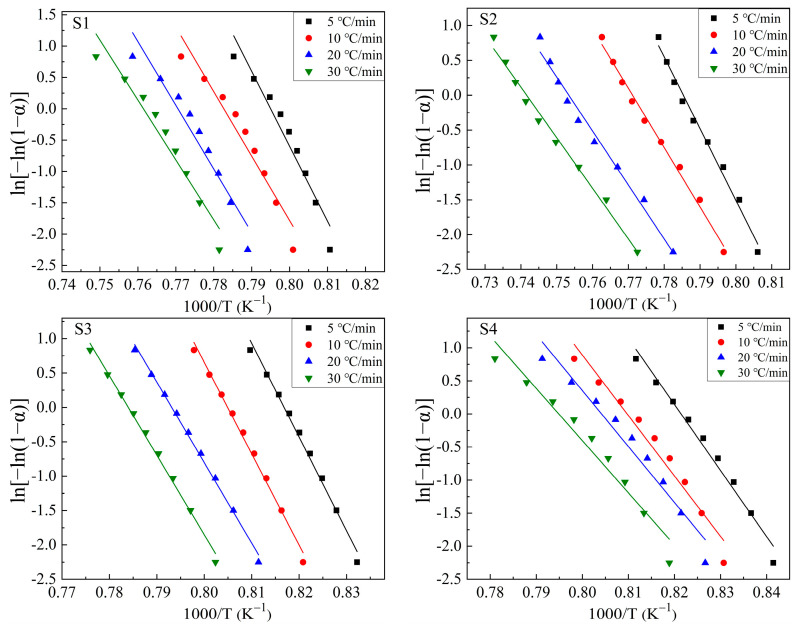
ln[−ln(1−*α*)]~1000/T at different heating rates for S1–S4 glass samples, to calculate the value of m*E*_G_.

**Figure 8 materials-16-04165-f008:**
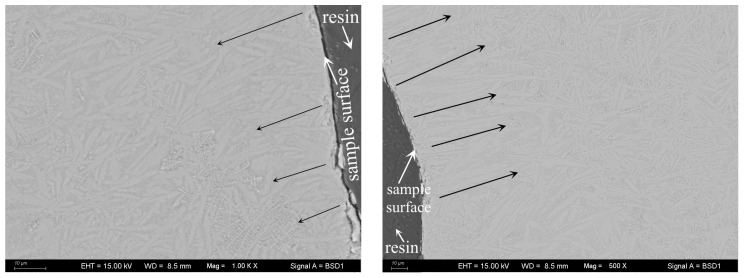
Observation of crystal growth mechanism for S4 glass via SEM-BSE (the two photographs are from different points of the same sample). The arrows are used to mark the growth direction of the crystals which are in light-grey color in the image, indicating that the columnar crystal grains form near the sample’s surface and are nearly perpendicular to the surface.

**Figure 9 materials-16-04165-f009:**
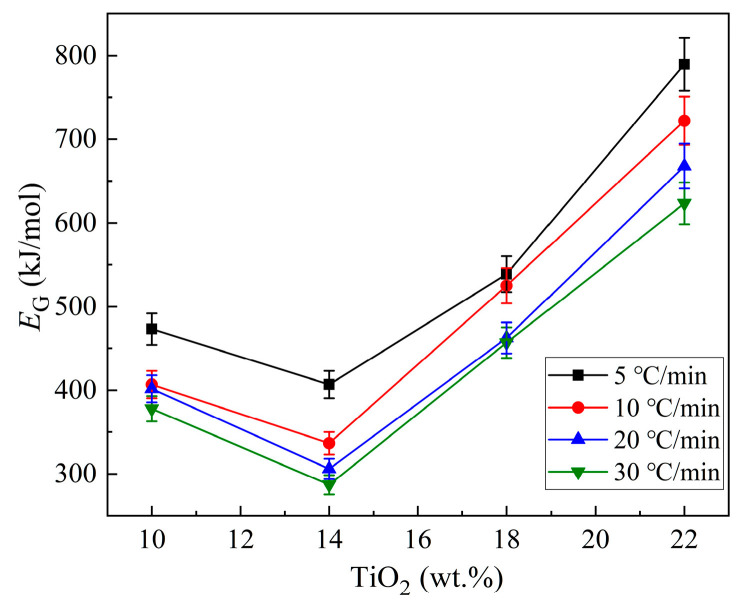
Changes in *E*_G_ with the increase in TiO_2_ in the CaO-SiO_2_-Al_2_O_3_-TiO_2_ fine-particle glass.

**Figure 10 materials-16-04165-f010:**
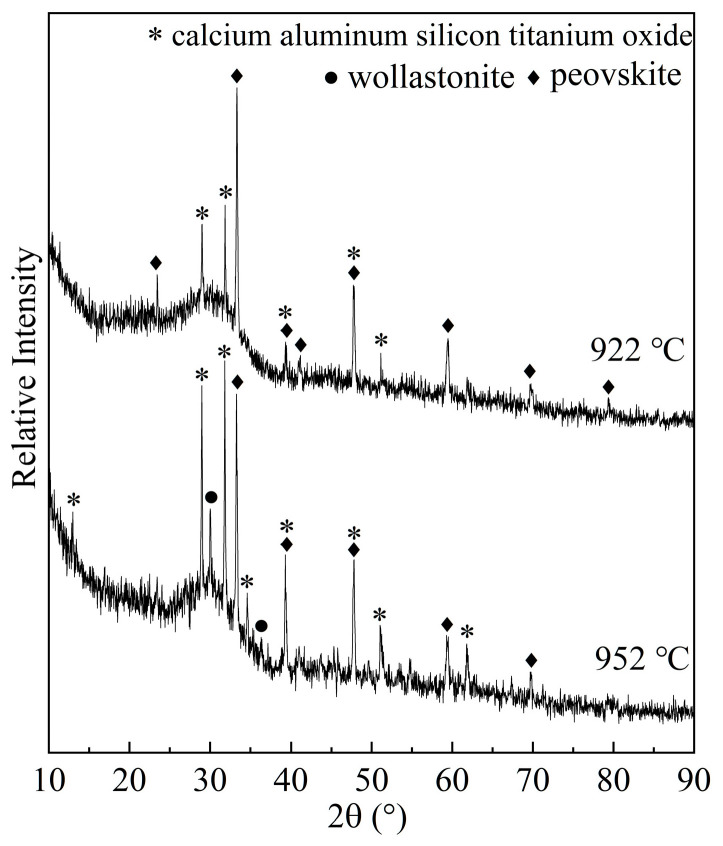
XRD patterns for the S3 sample crystallized at different temperatures to identify the sequence of crystallization of different phases.

**Table 1 materials-16-04165-t001:** Chemical composition (wt.%) of the studied glass.

Sample Number	CaO	SiO_2_	Al_2_O_3_	TiO_2_	w(CaO)/w(SiO_2_)
S1	40	40	10	10	1
S2	38	38	10	14	1
S3	36	36	10	18	1
S4	34	34	10	22	1

**Table 2 materials-16-04165-t002:** Parameters of the crystallization kinetics and the identified crystals of CaO-SiO_2_-Al_2_O_3_-TiO_2_ glass with fine particles.

Sample Number	Heating Rate	m*E_G_* (kJ/mol)	n	m	*E_G_* (kJ/mol)	Crystals
S1-10%TiO_2_	5 °C/min	946.11	2.14	2	473.06	CaSiO_3_, CaTiO_3_, Ca_3_TiSi_2_(AlSiTi)_3_O_14_
	10 °C/min	813.46	1.84	2	406.73	
	20 °C/min	803.26	1.82	2	401.63	
	30 °C/min	755.63	1.71	2	377.81	
S2-14%TiO_2_	5 °C/min	813.35	2.38	2	406.67	CaSiO_3_, CaTiO_3_, Ca_3_TiSi_2_(AlSiTi)_3_O_14_
	10 °C/min	673.20	1.97	2	336.60	
	20 °C/min	612.08	1.79	2	306.04	
	30 °C/min	573.88	1.68	2	286.94	
S3-18%TiO_2_	5 °C/min	1077.44	2.40	2	538.72	Ca_3_TiSi_2_(AlSiTi)_3_O_14_, CaTiO_3_, CaSiO_3_
	10 °C/min	1050.09	2.34	2	525.04	
	20 °C/min	924.80	2.06	2	462.40	
	30 °C/min	913.46	2.04	2	456.73	
S4-22%TiO_2_	5 °C/min	789.38	1.30	1	789.38	Ca_3_TiSi_2_(AlSiTi)_3_O_14_, CaTiO_3_, CaSiO_3_
	10 °C/min	722.00	1.19	1	722.00	
	20 °C/min	667.91	1.10	1	667.91	
	30 °C/min	623.36	1.03	1	623.36	

## Data Availability

The data presented in this study are available upon request from the corresponding author.

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
