# Peer review of "A Kinetic Study on Crystallization in TiO2-SiO2-CaO-Al2O3 Glass under Nucleation Saturation Conditions for the High Value-Added Utilization of CaO-SiO2-Based Solid Wastes"

_materials, 2023, doi:10.3390/ma16114165_

Round 1

Reviewer 1 Report

The work proposed for consideration considers as its main goal the consideration of the mechanisms of nucleation, phase formation and growth of crystals in a complex oxide system TiO2-SiO2-CaO-Al2O3 during the formation of glass ceramics.
The overall impression is that this work was done to a very high standard. This impression is due to the following circumstances.
- The introduction provides an exhaustive justification for the importance and relevance of the subject and object of study. The novelty and originality of the study is substantiated.
- The selected research methods (DTA, XRD, SEM, EDS) are fully consistent with the object of study. These methods complement each other properly and provide clear and unambiguous results.
- The results of the study are presented clearly, clearly and consistently. The results do not raise doubts or desire to request any clarifications.
- The conclusion is presented in an understandable way. All the conclusions made in the work are confirmed by the observed results and fully fit into the existing theory of kinetics and crystallization mechanisms.
The reliability of the work (in my humble opinion) is confirmed, for example, by the fact that the authors take into account greater kinetic difficulties in the formation of more chemically complex phases (Ca3TiSi2(AlSiTi)3O14) compared to simpler ones (CaTiO3 and CaSiO3). In my practice crystal growth in water solutions I find similar experimental evidence.
The novelty and value of the work lies, in my opinion, in the controlled factor of the grain dimension in the formation of ceramics. The approach used by the authors made it possible to level the uncertainties that arise when taking into account the number of crystallization centers in the kinetic model. Moreover, the proposed approach (in perspective may allow a more accurate estimate of the concentration of crystallization centers when conducting a similar study with initial particles of other sizes.
Summarizing, we can say that even if there is a desire to complain about something in this work, then there is nothing to complain about in it. I strongly recommend this work for publication in the form in which it is presented.

Reviewer 2 Report

The reviewed paper is interesting. However, I believe that before it can be accepted for publication in the journal Materials, it must first be rigorously revised with respect to the English wording, which is the main problem that I have perceived in the paper. In general, this wording is deficient, so that in some parts of the paper the text is very confusing, which I do not mention specifically because I consider it necessary to revise the entire document in this regard. Also, throughout the paper many erroneous expressions appear, such as "heating treat", however, they are too many to mention them all. Such revision should preferably be carried out with the help of a person whose native language is English. Other aspects of the paper that should also be addressed are the following:

- In the appropriate places of both the Abstract and the Introduction sections of the paper, it is necessary to define expressions such as "Site Saturation" and “EG”, since suddenly the authors start mentioning them without having defined them before.

- Line 95 on page 2 mentions some equations, but those equations are not given there.

- In lines 104 and 105 on page 3 it is mentioned that when considering the "Site Saturation" phenomenon, it is necessary to keep in focus the effects of the size of the glass particles for the correct calculation of the kinetic parameters for the devitrification process. However, it is not explained what these effects are. Presumably the authors refer to the effects of particle size on the nucleation and growth of the crystalline phases that take place on the surface of the glass particles, during the devitrification of the latter. However, perhaps in such a case it is more appropriate to consider the surface area of the glass particles rather than their size. Please discuss this in the paper.

- Verify if there is a more appropriate name for the "calcium aluminum silicon titanium oxide [Ca 3TiSi2(AlSiTi)3O14]" mentioned in several parts of the paper.

- There is a spelling error inside Figure 2.

- Please arrange Figures 3 and 6 so that each of them is complete, along with its Figure footer, on a single page.

- In lines 193 and 194 on page 6, on what basis was it decided that a particle size of 58 mm was the dividing boundary between large particles and fine particles? What was the dispersion (standard deviation) for the fine-sized particles used? It only says that fine particles have a size below 58 mm, but what was the smallest particle size used?

- How was the volumetric fraction (α) of the phase formed determined from the crystallization peaks of the DTA curves (line 218, page 7)?

- Apparently Equations 2 and 3 of the paper were originally derived based on different physical assumptions, by different authors. So, do they both produce the same values for the mEG/R ratio, keeping all conditions constant? Please discuss this in the paper.

- How does the value of α was kept constant at certain values to obtain the lines shown in Figure 6? Specify this in the paper.

- In lines 305 to 307 on page 11 it says that equation 1 is derived based on the assumption that a sufficiently large number of nuclei already existed in the as-quenched glass before the DTA tests were carried out. What is this assumption based on? Where do these nuclei come from? What is the nature of these nuclei? Please discuss this in the paper.

- Finally, please check the format of all references to verify that it is correct. For example, in Ref. 26, line 496 on page 15, there is a weird expression in the name of the last co-author.

See above

Reviewer 3 Report

This version does not look worthy and cannot be recommended for publication in this form and at least needs major revision.

1.     EG needs to be explained in the Abstract.

2.     Line 32. This sentence needs supporting reverences, confirming novelty and relevance. Recently, such and similar glasses were investigated in details for other different applications, as shielding, solid oxide fuel cell, luminescent materials etc

Kozlovskiy, A., et al (2022). Investigation of the Efficiency of Shielding Gamma and Electron Radiation Using Glasses Based on TeO2-WO3-Bi2O3-MoO3-SiO to Protect Electronic Circuits from the Negative Effects of Ionizing Radiation. Materials15(17), 6071. https://doi.org/10.3390/ma15176071

Saetova, N.S., et al. Alumina–silica glass–ceramic sealants for tubular solid oxide fuel cells. J Mater Sci 54, 4532–4545 (2019). https://doi.org/10.1007/s10853-018-3181-8

Antuzevics, A., et al (2018). Crystalline phase detection in glass ceramics by EPR spectroscopy. Low Temperature Physics44(4), 341-345.

3.     How important are point defects in these ceramics? In the introduction the problems of work are insufficiently considered. It is unlikely that ceramics are created without structural oxygen vacancies (F –type centers), but the latter always play a very important role in the constituent oxides.

4.     What is somehow completely unclear in the Introduction, what is the role and importance of understanding pores and shaping porosity?

5.     Paragraph 2.1.  There is not enough information about starting materials and their structures.

6.     Figure 5. Give more detailed information about the given spectra and explain all these observed features.

7.     Table 2. These data need error bars and corresponding comments in the text.

8.     Table 2. To confirm the presence of all the mentioned crystals, it would be important to measure the Raman spectra.

Round 2

Reviewer 2 Report

In general, the corrections I suggested to make to the first version of the paper have been made satisfactorily, so I have no objection to recommending that the corrected version of it be accepted for publication in the journal Materials.

There are still some small corrections regarding the quality of the English language that need to be made in the paper. Please review the wording of the paper again.

Reviewer 3 Report

I am absolutely dissatisfied with the answer of the authors to question 8:

Author reply: 8. Table 2. To confirm the presence of all the mentioned crystals, it would be important to measure the Raman spectra. Response: The presence of all the mentioned crystals have been confirmed by the combination analysis of XRD and SEM results. XRD and SEM measurements are the classical methods to confirm the presence of crystals. Raman spectra can give some structure information, but the roles of Raman spectra in confirming the crystal are limited.

I would like to note that Light elements (N<14) are difficult to identify by SEM, so Raman is necessary because it also allows to determine the role of hydrogen, OH, H2O.

Furthermore, hydrogen cannot be seen in XRD, therefore, all products of interaction with hydrogen cannot be properly identified.   Rejection of complementary method is serious drawback of this paper.

Photoluminescence is also one of the complementary methods.

In fact, Fig.2 contains, as it were, information about the presence of several different components, but a detailed analysis has not been done. It is assumed that the presence of separate coinciding points already proves what the authors would like to claim.
But this is not at all obvious at this stage. Therefore, I wanted to see evidence of the presence of all these  components from corresponding photoluminescence and Raman spectra measurements.
